



# Observer-based power forecast of individual and aggregated offshore wind turbines

Frauke Theuer[1], Andreas Rott[1], Jörge Schneemann[1], Lueder von Bremen[2], and Martin Kühn[1]

[1]ForWind, Institute of Physics, University of Oldenburg, Küpkersweg 70, 26129 Oldenburg, Germany
[2]German Aerospace Center (DLR) - Institute of Networked Energy Systems, Carl-von-Ossietzky-Straße 15, 26129 Oldenburg

**Correspondence:** Frauke Theuer (frauke.theuer@uni-oldenburg.de)

**Abstract.** Due to the increasing share of wind energy in the power system, minute-scale wind power forecasts have gained importance. Remote sensing-based approaches have proven to be a promising alternative to statistical methods and thus need to be further developed towards an operational use, aiming to increase their forecast availability and skill. Therefore, the contribution of this paper is to extend lidar-based forecasts to a methodology for observer-based probabilistic power forecasts of individual wind turbines and aggregated wind farm power. To do so, lidar-based forecasts are combined with SCADA-based forecasts that advect wind vectors derived from wind turbine operational data. After a calibration, forecasts of individual turbines are aggregated to a probabilistic power forecast of turbine subsets by means of a copula approach. We found that combining the lidar- and SCADA-based forecasts significantly improved both forecast skill and forecast availability of a 5-minute ahead probabilistic power forecast at an offshore wind farm. Calibration further increased the forecast skill. Calibrated observer-based forecasts outperformed the benchmark persistence for unstable atmospheric conditions. The aggregation of probabilistic forecasts of turbine subsets revealed the potential of the copula approach. We discuss the skill, robustness and dependency on atmospheric conditions of the individual forecasts, the value of the observer-based forecast, its calibration and aggregation and more generally the value of minute-scale power forecasts of offshore wind. In conclusion, combining different data sources to an observer-based forecast is beneficial in all regarded cases. For an operational use one should distinguish between and adapt to atmospheric stability.

## 1 Introduction

With the increasing share of wind and solar power in our energy system, the need for accurate minute-scale power forecasts to support grid stability and electricity trading arises (Dowell and Pinson, 2016; Sweeney et al., 2020; Würth et al., 2019). The low geographical dispersion of installed offshore wind capacity and its consequently high volatility (Malvaldi et al., 2017) calls for skillful forecasts of in particular offshore wind power. Commonly, statistical methods, such as the benchmark persistence or AR(I)MA (Auto-Regressive (Intergrated) Moving Average) methods, are applied on those time scales (Würth et al., 2019). While those methods are reliable in many situations, they underperform, for instance, during ramp events, i.e. sudden and strong changes in wind speed or direction. Therefore, recently remote sensing-based wind speed and power forecasts have





been researched as a physical-based alternative (Würth et al., 2018; Valldecabres et al., 2018b, a, 2020; Theuer et al., 2020a,
2021; Pichault et al., 2021).

Several studies have shown the potential of lidar-based wind speed and power forecasts to outperform the benchmark persistence under specific atmospheric conditions (Valldecabres et al., 2018b; Theuer et al., 2021; Pichault et al., 2021). Theuer et al. (2020b) and Valldecabres et al. (2018b) found that atmospheric stability can influence forecast accuracy in particular with respect to the wind speed height extrapolation. Theuer et al. (2021) showed that overall lidar-based forecasts are more
accurate during stable conditions, however, they can only outperform persistence during unstable stratification because also persistence is more skillful during stable situations. Valldecabres et al. (2020) introduced a dual Doppler radar-based forecast that was able to outperform persistence in terms of probabilistic scores during ramp events and for free stream turbines. Two lidar-based methods, one based on a neural network and one on a smart persistence approach, introduced by Pichault et al. (2021) were able to exceed persistence as well as an ARIMA method during ramp events and non-ramp situations, for different
wind directions and atmospheric conditions onshore. In their work the authors focus on deterministic forecasts and wind farm power forecasts that do not distinguish forecasts at turbine level.

Driven by these promising results, the methods' development now needs to be directed towards an operational use. Besides the fact that there are many situations during which persistence outperforms the lidar-based forecast, low forecast availability is a main issue with the technology and concepts available so far. Hence, depending on the wind farm layout, scanning trajectories,
lidar availability and wind conditions, no or only low-quality forecasts can be generated (Theuer et al., 2020b). This problem can be reduced by optimizing scanning trajectories, increasing the lidar's measurement range and possibly commissioning additional devices. However, during situations with reduced lidar sight, due to e. g. fog or rain, or when devices fail, one would need to fall back to an alternative data source. For that purpose, hybrid methods are worth to be considered. In the context of lidar-based methods, Theuer et al. (2021), for instance, showed that the additional use of wind turbine operational data can
contribute to the forecast accuracy. Also Pichault et al. (2021) included wind farm operational data in the form of a smart persistence approach in their forecast and achieved promising results.

Currently, lidar-based methods have been evaluated with regard to their probabilistic characteristics in a few cases only (Theuer et al., 2020b) but mainly with respect to their deterministic characteristics and for individual wind turbines (Würth et al., 2018; Valldecabres et al., 2018b; Theuer et al., 2021). However, for end-users in power trading and system operation,
uncertainty information is of high value as it aids decision making processes (Dowell and Pinson, 2016; Sweeney et al., 2020). One way to increase the reliability and sharpness of probabilistic forecasts is statistical post-processing, i. e. forecast calibration (Thorarinsdottir and Gneiting, 2010). Commonly, ensemble model output statistics (EMOS) is used. EMOS was first developed for temperature and pressure forecasts (Gneiting et al., 2005) but has successfully been applied to the prediction of precipitation (Scheuerer, 2014), wind speed (Thorarinsdottir and Gneiting, 2010), wind vectors (Schuhen et al., 2012) and
power (Späth et al., 2015).

Considering the different areas of application of minute-scale forecasts, both individual turbines' power output and aggregated wind farm power or power at the grid connection point, i. e. aggregated power of a subset of individual wind turbines, are important. While the former are mainly required for wind turbine control (Würth et al., 2019), the latter are of interest





for trading and system operation purposes. So far, lidar-based forecasts of individual wind turbines focused on free-stream

situations. In a next step, these methodologies need to be extended to wake-influenced turbines. A main challenge is hereby the propagation technique, which assumes constant wind vector trajectories and is therefore unable to account for wakes. Valldecabres et al. (2020) circumvent this by applying a directional turbine efficiency that significantly improved the skill of their radar-based forecast.

Individual turbines' power forecasts can also be helpful when determining wind farm power. In this context, recently hi-
erarchical forecasting on both temporal as well as spatial levels has gained attention, aiming to achieve coherency between different levels of the hierarchy and thereby improving forecast performance at each level (Bessa, 2016; Gilbert et al., 2020). A common method in the context of coherent probabilistic forecasts are copula approaches. Gilbert et al. (2020) successfully implemented and tested a variety of copulas to aggregate the probabilistic power forecasts of individual wind turbines to the probabilistic forecast of wind farm power.

Our objective in this paper is to develop a probabilistic observer-based forecast of aggregated wind farm power. To do so, we first introduce an observer-based power forecast of individual wind turbines that combines lidar and turbine operational data. This method accounts for variable wake conditions and increases forecast availability and skill. Additional calibration further improves the forecast's probabilistic characteristics. In the second step, we aggregate individual probabilistic wind turbine power forecasts to probabilistic wind farm power forecasts by applying a copula approach.

## 2  Methods

The basis of this work is the lidar-based forecasting approach introduced and analysed in more detail in Theuer et al. (2020a), Theuer et al. (2020b) and Theuer et al. (2021). The method is briefly described in Section 2.1. In this work this approach is significantly extended further as described in the following Section. Using SCADA (Supervisory Control and Data Acquisition) data, it is first extended to an observer-based forecast (OF) to increase forecast availability and skill (cf. Section 2.2). In a next
step, observer-based forecasts are calibrated by means of Ensemble Model Output Statistics (EMOS) (cf. Section 2.3). Finally, probabilistic power forecasts of individual wind turbines are aggregated using different copula approaches (cf. Section 2.4).

### 2.1  Reference method lidar-based forecast (LF)

Lidar-based power forecasts (LF) (Theuer et al., 2020a) utilize horizontal or slightly elevated plan position indicator (PPI) lidar scans measuring the inflow of an offshore wind farm. Typically, lidar devices are positioned on the transition piece (TP) of a
wind turbine or alternatively a nearby platform and record line-of-sight (LOS) wind speed measurements and the carrier-to-noise ratio (CNR) at each scanned azimuth angle and range gate along with a time stamp. Using that information, lidar scans are filtered applying a data density approach on normalized CNR values and LOS wind speed measurements similar to Beck and Kühn (2017). By means of a VAD-like fit, the wind direction $\chi$ is then determined dependent on range gate $r$ (Werner, 2005) and used to reconstruct a wind field with the horizontal wind speed $u_{\mathrm{h}}$ from the line-of-sight wind speed measurements





$u_{\text{LOS}}$ and the lidar's azimuth angle $\vartheta$

$$u_{\text{h}}(r,\vartheta) = \frac{u_{\text{LOS}}(r,\vartheta)}{\cos(\vartheta - \chi(r))}. \tag{1}$$

After wind field reconstruction, the individual lidar scans are interpolated to a cartesian grid and synchronized in time to account for the large time shift within each scan (Beck and Kühn, 2019). A Lagrangian advection technique is then applied to propagate wind vectors, i. e. horizontal wind speed and wind direction information at each grid point. Hereby, it is assumed that vectors

travel with their local wind speed and wind direction and do not change their trajectory while travelling. Wind vectors reaching the area of influence around the target turbine within a time interval of $k \pm 30\,\text{s}$ with lead time $k$ are selected to contribute to the target turbine's probabilistic forecast. For each forecasted time step, wind data recorded during a time interval previous to forecast initialization is taken into account. That means, for each forecast several time-synchronized scans are considered and the travelling time of wind vectors can therefore exceed the lead time. Considering also previous scans is important to be

able to forecast turbines positioned further away from the lidar-scanned area. Wind speed forecasts at measurement height $u_{\text{m}}$ are transformed to hub height assuming a logarithmic stability corrected wind speed profile (Emeis, 2018). Here, we apply a methodology introduced as tendency-based forecast in previous work (Theuer et al., 2021). It determines the wind speed tendency at measuring height and applies it to wind speed at hub height $u_{\text{hh}}$ after performing a correction of measuring height $z_{\text{m}}$ and atmospheric conditions defined by the Obukhov length $L$ and the roughness length $z_0$ between time steps $t_i$ and $t_{i-1}$ (cf.

Equation 2). $\Psi(z,L)$ describes the stability correction term (Emeis, 2018). Measuring heights vary along the range gate due to the curvature of the Earth and dynamically due to a thrust-dependent tilt of the lidar device (Rott et al., 2022). The hub height wind speed at the future time step $t_i$ is then defined as

$$u_{\text{hh}}(t_i) = \frac{\ln\left(\frac{z_{\text{m}}(t_{i-1})}{z_0(t_i)}\right) - \Psi\left(\frac{z_{\text{m}}(t_{i-1})}{L(t_i)}\right)}{\ln\left(\frac{z_{\text{m}}(t_i)}{z_0(t_i)}\right) - \Psi\left(\frac{z_{\text{m}}(t_i)}{L(t_i)}\right)} \frac{u_{\text{m}}(t_i)}{u_{\text{m}}(t_{i-1})} u_{\text{hh}}(t_{i-1}). \tag{2}$$

In a final step, the wind speed forecast is transformed to a power forecast using power curves extracted individually for each

wind turbine from 1-minute-mean SCADA wind speed and power data. In this case, the wind speed values are not measured but estimated from power, pitch angle and the SCADA system's turbine power curve.

Details on this forecasting methodology can be found in Theuer et al. (2020b) and Theuer et al. (2021).

## 2.2 Extension to an observer-based forecast (OF) by integrating a SCADA-based forecast (SF)

If the LF is invalid due to missing data, the prevailing wind conditions or the lidar trajectory or wind farm layout one needs

to fall back to an alternative forecasting approach. For that purpose we introduce the observer-based forecast, which combines the LF and a SCADA-based forecasting approach.

The SCADA-based power forecast (SF) modifies the methodology introduced in Rott et al. (2020). The 1-Hz wind speed and wind direction data of all wind turbines of the wind farm are propagated using Lagrangian advection. In accordance with





the LF, only wind vectors $v$ arriving within a certain area of influence around our target turbine $j$ are selected. The selected

vectors originating at time $t_{v,j}$ are then weighted according to their age $t - t_{v,j}$ using an inverse temporal distance weighting to determine the weighting factor $\hat{w}_{v,j}(t)$

$$\hat{w}_{v,j}(t) = \frac{w_{v,j}(t)}{\sum_v w_{v,j}(t)}, \tag{3}$$

with

$$w_{v,j}(t) = \frac{1}{(t - t_{v,j})^p} \tag{4}$$

and the tuning parameter $p \in \mathbb{N}$ that determines the strength of the weighting factor's decrease with increasing temporal distance. The selected wind vectors are resampled to a predefined number of wind vectors with their individual contribution given by the weighting factor. As suggested by Rott et al. (2020) a bias correction with the observed wind speed $u_{\mathrm{obs},j}$ and the ensemble average of the forecast at turbine $j$, i. e. $\bar{u}_{\mathrm{sc},j}$, is applied to all members $v$ of the forecast at this turbine $u_{\mathrm{sc},v,j}$ to account for possible systematic errors and wake effects. The bias-corrected wind speed vectors $u_{\mathrm{corr},v,j}$ then yield

$$u_{\mathrm{corr},v,j}(t) = u_{\mathrm{sc},v,j}(t) - \left( \frac{1}{N_{\mathrm{t}}} \sum_{l=0}^{N_{\mathrm{t}}} \bar{u}_{\mathrm{sc},j}(t - k - l\,\Delta\tau) - u_{\mathrm{obs},j}(t - k - l\,\Delta\tau) \right), \tag{5}$$

with $N_{\mathrm{t}}$ the number of time steps with length $\Delta\tau$ prior to forecast initialization $t - k$, with lead time $k$, considered to determine the bias. Wind speed forecasts are transformed to power forecasts as described for the LF (Section 2.1).

If both LF and SF are valid, they are weighted equally in the OF; otherwise only the valid forecast is considered. To be considered valid we require a minimum number of wind vectors to reach the target turbine for both methods. That way,

we avoid individual wind vector outliers being given too much weight. To account for the varying number of wind vectors contributing as a consequence of different temporal and spatial resolutions of the lidar and SCADA data, we resample each forecast to contain the same predefined number of members.

### 2.3 Calibration of the observer-based forecast

In a next step, the OF is calibrated using Ensemble Model Output Statistics (EMOS). Hereby, a truncated Gaussian distribution


$$f(x, \mu, \sigma) = \frac{1}{\sigma} \frac{\phi\left(\frac{x-\mu}{\sigma}\right)}{\Phi\left(\frac{P_{\mathrm{r}}-\mu}{\sigma}\right) - \Phi\left(\frac{0-\mu}{\sigma}\right)} \tag{6}$$

for $0 \leq x \leq P_{\mathrm{r}}$ and $f(x < 0) = 0$ and $f(x > P_{\mathrm{r}}) = 0$ with rated power $P_{\mathrm{r}}$ is used to model the wind speed distribution (Thorarinsdottir and Gneiting, 2010). The probability density function of the standard normal distribution is defined by $\phi$ and its



cumulative distribution function (cdf) by $\Phi$. The mean $\mu_{i,j}$

$$\mu_{i,j} = a + b\,\overline{\text{fc}}_{i,j} \tag{7}$$

and the variance $\sigma_{i,j}^2$ of the distribution

$$\sigma_{i,j}^2 = c + d\,\text{fc}_{\sigma^2 i,j} \tag{8}$$

are modelled as a linear function of the ensemble mean $\overline{\text{fc}}_{i,j}$ respectively variance $\text{fc}_{\sigma^2 i,j}$ with time index $i$ and turbine index $j$. The cdf of the ensemble members at time $i$ and for turbine $j$ is defined as $F_{i,j}(\mu_{i,j}(a,b),\sigma_{i,j}(c,d))$ and referred to as $F_{i,j}$ in the following. The parameters $a$, $b$, $c$ and $d$ are optimized to minimize the cost function

$$J_j(x_{i,j},a,b,c,d) = \frac{1}{N_{\text{c}}} \sum_{i=1}^{N_{\text{c}}} \text{crps}(F_{i,j},x_{i,j}) \tag{9}$$

based on the continuous ranked probability score (crps) of the forecast

$$\text{crps}(F_{i,j},x_{i,j}) = \int_0^{P_{\text{r}}} [F_{i,j}(x) - H(x - x_{i,j})]^2 dx. \tag{10}$$

with the observation $x_{i,j}$ the number of time steps considered $N_{\text{c}}$ and the Heaviside step function $H$ (Gneiting et al., 2007). A sliding window approach is applied, thus a training interval with optimized length before forecast initialization is used to calibrate the forecast.

## 2.4 Aggregated wind turbine power forecast using a copula approach

The observer-based forecast provides probabilistic power forecasts of individual wind turbines, i. e. one cdf $F_{i,j}$ for each time index $i$ and individual wind turbine $j$. Here, we aim to derive a joint predictive distribution of wind power production from a subset of wind turbines in a wind farm using a copula approach. This approach is based on Sklar's theorem, which states that a $m$-dimensional cumulative distribution $F$, with the number of turbines $m$ and the length of the training data set $t_{\text{n}}$, can be expressed using a copula function $C$ of the individual marginal distributions $F_{i,j}$ as

$$F(x_{1,1},x_{1,2},...,x_{t_{\text{n}},m}) = C(F_{1,1}(x_{1,1}),F_{1,2}(x_{1,2}),...,F_{t_{\text{n}},m}(x_{t_{\text{n}},m})), \tag{11}$$

conditional on well-calibrated forecasts with uniformly distributed marginals $u_j = F_j(x_j)$ (Gilbert et al., 2020). In this work, we apply a Gaussian copula

$$C(F_{1,1}(x_{1,1}),F_{1,2}(x_{1,2}),...,F_{t_{\text{n}},m}(x_{t_{\text{n}},m})) = \Phi_\Sigma(\Phi^{-1}(F_{1,1}(x_{1,1})),\Phi^{-1}(F_{1,2}(x_{1,2})),...,\Phi^{-1}(F_{t_{\text{n}},m}(x_{t_{\text{n}},m}))) \tag{12}$$



with the $m$-dimensional normal distribution $\Phi_\Sigma$ with covariance matrix $\Sigma$ and a mean of $\mu_1 = \mu_2 = ... = \mu_m = 0$. To determine the joint predictive distribution of the individual turbines and finally the probabilistic aggregated power, we proceed as follows: First, marginal distributions of all wind turbines to be considered for the aggregation are determined from the cdfs and observations as $F_{i,j}(x_{i,j})$ and their uniformity is verified (Pinson et al., 2009). Marginals are then transformed into the Gaussian domain described by $\Phi^{-1}(F_{i,j}(x_{i,j}))$. Based on these transformed and normally distributed marginals, the covariance matrix $\Sigma$ of the training data set can be determined. This multivariate distribution can be used to generate $M$ random samples, which are then transformed back to the uniform domain. Finally, for each turbine $j$ and time steps within the test data set $i$ the samples are transformed into the power domain using its cdf $F_{i,j}$ and summed over all turbines to yield a set of aggregated power samples. Based on these $M$ aggregated power samples, a power distribution, i. e. a probabilistic forecast can be derived.

To enlarge the test data set, we estimate covariance matrices using a sliding windows approach. This also allows us to determine a joint predictive distribution that flexibly adapts to changing atmospheric conditions. A change in wind direction, for example, will affect the wake situation of the turbines and is consequently expected to have an impact on the turbine subset's joint distribution too.

In addition to the empirical covariance determined as described above, we define and test parametric covariance matrices based on an exponential relation

$$\Sigma_{j,h} = \exp\left(-\frac{\Delta r_{j,h}}{\nu}\right) \tag{13}$$

with the covariance between two turbines $\Sigma_{j,h}$, and the spatial distance $\Delta r$ between the position of turbines $j$ and $h$ (Gilbert et al., 2020). The parameter $\nu$ is fitted using a least-squares regression and the empirically determined covariance matrix. The advantage of parametric copulas is their lower sensitivity to reduced data availability, avoiding noisy covariances and overfitting (Gilbert et al., 2020).

We further evaluate vine copulas as a more flexible option compared to Gaussian copulas. Vine copulas describe a set of bivariate copulas with variable distribution families for each (turbine) pair (Bessa, 2016). Here, we determined vine copulas using the Matlab framework developed by Coblenz (2021). Distribution families are chosen using the Akaike Information Criteria (AIC) (Aas et al., 2009).

## 3 Results

After the general description of the methodological steps in the previous section, we introduce the case study analyzed in this work and its case-specific parameters in Section 3.1. In Section 3.2 the results of the LF and SF for individual wind turbines are presented. Further, we assess the value of the OF compared to the LF, SF and persistence (Section 3.3) and evaluate the calibrated OF compared to the raw, i. e. the uncalibrated, one (Section 3.4). Finally, we determine the forecast skill of the aggregated probabilistic power of several wind turbines and compare it against a probabilistic version of persistence (Section 3.5).

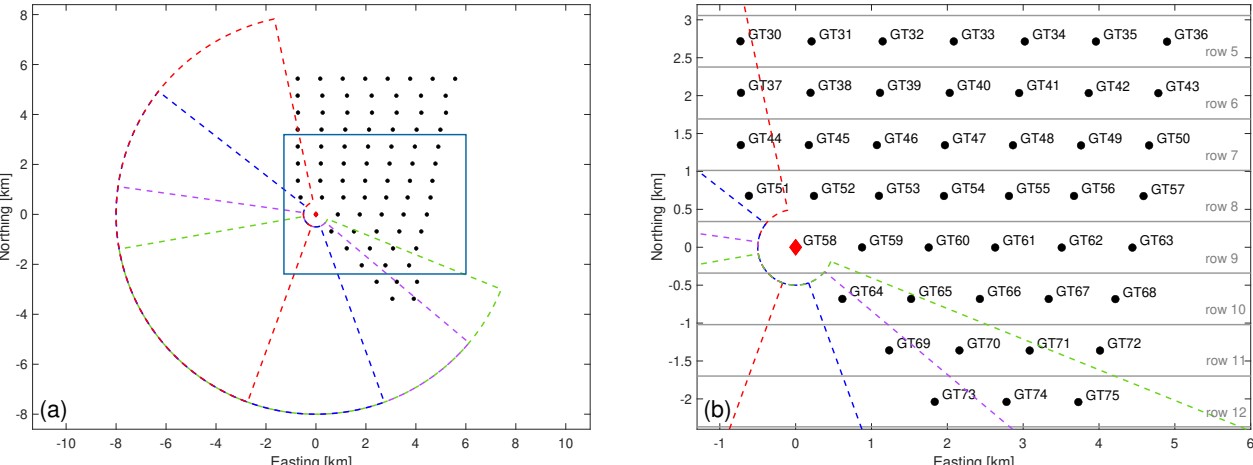

**Figure 1.** Layout of the wind farm Global Tech I with turbine positions visualized as black dots. The lidar location is depicted as red diamond and lidar trajectories as coloured dashed lines. The Cartesian grid is centered around the lidar's position. Grey horizontal lines mark turbine rows referred to in this work. The blue rectangle indicates the zoomed-in region shown on the right. It displays turbine numbers of the wind farm's centre region.

### 3.1 Case study at the offshore wind farm Global Tech I (GT I)

The methodology described in the previous sections is applied to and evaluated at the offshore wind farm Global Tech I (GT I)
in the German North Sea. The wind farm consists of 80 turbines of type Adwen AD 5-116, with a hub height of $z_{hh} = 92$ m, a
rotor diameter of $D = 116$ m and a rated power of $P_r = 5$ MW. The lidar was placed on the transition piece of turbine GT58 at
a height of $z_{TP} = 24.6$ m. Horizontal plan position indicator (PPI) lidar scans were performed with a WindCube 200S (Serial
no. WLS200S-024) and with an elevation of $0°$, an azimuth angle spanning $150°$, an azimuthal resolution of $2°$, range gates
from 500 m to 7950 m in 35 m intervals and an accumulation time of 2 s. Including the measurement reset time, the scanning
duration was 156 s. The scanning trajectories, which were adjusted manually according to four wind direction sectors, and the
wind farm layout are depicted in Figure 1 (b). More details on the measurement campaign are available in Schneemann et al.
(2020), Theuer et al. (2020b) and Theuer et al. (2021).

Each forecasted time step of the LF considered the six most recent scans, thus can contain wind data measured during the last
15 minutes. Wind vectors contributing to the SF were weighted using a tuning parameter of $p = 4$. The SF's bias correction
was performed considering a number of $N_t = 5$ time steps prior to forecast initialization. The step length was chosen as
$\Delta\tau = 156$ s in accordance with that of the lidar scans. LF and SF were generated with an area of influence of $2D$ and a
minimum of 20 required wind vectors and were resampled to contain 500 members. Forecast calibration was performed with a
5 h training interval before forecast initialization. The time window was optimized in a sensitivity analysis. A calibration was
only performed for situations with at least 60 % valid data within that training period.





To construct a joint predictive distribution of all turbines of GT I a sufficiently large training data set with simultaneously
available forecasts of all turbines is required. As a consequence of the limited forecast availability, we therefore only considered
subsets of turbines to generate and evaluate aggregated power forecasts in this work. Turbine subsets were selected based on
the availability of simultaneously available forecasts and their proximity to each other (cf. Figure 1 (b)). Here, a 6 h training
window was used, again determined using a sensitivity analysis.

For forecast calibration, training of the copula and forecast evaluation 1-Hz SCADA power data, averaged to 1 minute
intervals, was used.

### 3.2    Evaluation of lidar-based and SCADA-based power forecasts for individual wind turbines

We evaluate 5-minute-ahead power forecasts generated within the period 8 March 2019 to 21 June 2019 against 1-minute-mean
SCADA data. In total, 9438 valid forecasts were generated and 6753 were successfully calibrated. Hereby, we considered only
situations during which both lidar and SCADA data were available for forecast generation and evaluation and persistence
forecasts were available as a reference. The benchmark persistence assumes the future value equals the current observation. A
probabilistic version of persistence was constructed by adding forecasting errors of the past 19 time steps to the current forecast
as described by Gneiting et al. (2007). Further, forecasts of individual turbines not in normal operation mode were neglected.
The wind conditions of the 9438 analyzed time steps are summarized as a wind rose in Figure 2. Wind speed and wind direction
were extracted from the horizontal PPI lidar scans. The Obukhov length $L$ reaches values as small as -27 m in unstable and
11 m in stable cases. Median values of $L$ are -266 m for $L < 0$ and 268 m for $L > 0$, respectively. In the following analysis we
will distinguish between stable ($L > 0$) and unstable ($L < 0$) atmospheric conditions in accordance with the definition of the
stability-corrected logarithmic wind speed profile.

The forecast skill was determined by means of the average continuous ranked probability score

$$\overline{\text{crps}} = \frac{1}{N} \sum_{i=1}^{N} \text{crps}_i. \tag{14}$$

To compare the skill of two forecasts the crps skill score (crps ss)

$$\text{crps ss} = 100 \left( 1 - \frac{\overline{\text{crps}}}{\overline{\text{crps}}_{\text{ref}}} \right) \tag{15}$$

with the reference forecast $\overline{\text{crps}}_{\text{ref}}$ is applied.

To understand the impact of lidar coverage and turbine location on the forecast skill and forecast availability of LF and SF
we depict the number of available forecasts for each method in Figure 3 (a) and (b). In Figure 4 we further compare the crps
ss of the LF and SF with persistence as reference for individual turbines of GT I and distinguish between unstable and stable
atmospheric conditions. Based on the number of available forecasts the turbines GT30-GT75 (cf. Figure 1) were selected for
further analysis. Grey vertical lines mark horizontal wind turbine rows, with the turbine left of the line located on the easterly
side of the wind farm.



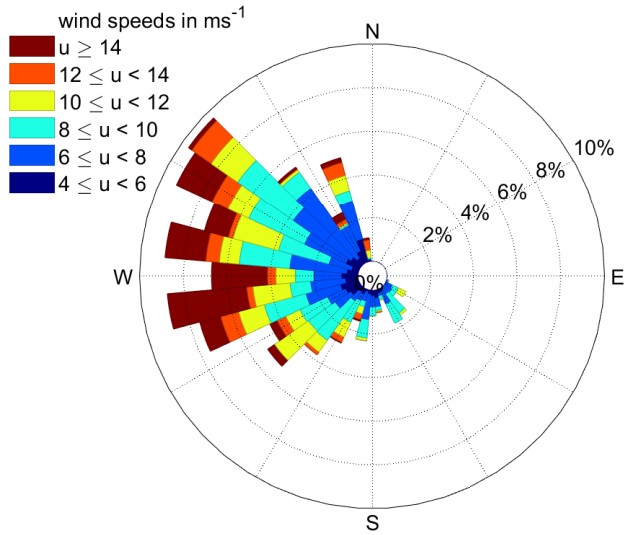

**Figure 2.** Wind speed and wind direction distribution extracted from horizontal PPI lidar scans of the 9438 analyzed time steps.

The westerly corner of the wind farm shows high LF availability (cf. Figure 2). In agreement with this, the LF was able to outperform persistence during unstable atmospheric conditions for those turbines covered well by the lidar scans (e. g. GT52, GT58, GT64). Its forecast availability is reduced for turbines located further away from the lidar. Here, also the forecast skill is low. This can be attributed to the longer time and distance wind vectors need to travel before reaching these turbines. Even though we consider in addition to the current lidar scan also previous ones, missing or low-quality scans increase the risk of wind vectors not reaching the turbines and negatively impact forecast skill. Moreover, high uncertainty might be related to wake effects. Wind turbines located in the northerly region of the wind farm show a low skill score due to insufficient lidar coverage. The SF mainly covers the easterly part of the wind farm and consequently performs well for easterly located turbines (e. g. GT50, GT57, GT63, cf. Figure 4), also during unstable conditions. It cannot predict free-flow turbines, considering the main westerly wind direction, as no upstream turbines are available to propagate from. Hence, skill scores are lower for turbines positioned close to the first row. Overall, the results indicate that both methods are able to predict power of not only free-stream turbines but also wake-influenced turbines more accurately than persistence under unstable conditions. During stable stratification both methods fail, in particular the SF.

Other than the SF, the LF is not bias-corrected to account for systematic errors possibly related to wakes. We therefore consider it worthwhile to analyze the impact of wakes on the LF in more detail. To do so, the $\overline{\mathrm{crps}}$ and the bias of GT30-GT75 are depicted in Figure 5 for wind directions $260° - 280°$ (a,b) and $170° - 190°$ (c,d). To capture in particular situations strongly impacted by wakes, we included only stable atmospheric conditions and situations operating below rated power ($< 0.9 P_r$) in this analysis. The $\overline{\mathrm{crps}}$ deteriorates, i. e. is growing, with increasing distance to the free-stream turbines. In accordance with the wind directions, forecasts are most accurate for westerly located turbines in Figure 5 (a) and for southerly located ones, with the exception of GT75, in (c). The bias is not distinctly affected by the individual turbines' position in the wind farm and



**Figure 3.** Forecast availability for (a) the lidar-based forecast, (b) the SCADA-based forecast, (c) the observer-based forecast and (d) the observer-based forecast after filtering situations during non-normal operation. The subset of turbines shown in (d) is analyzed in more detail in this work. The colour scale and magnitude of the dots visualize the number of valid forecasts.

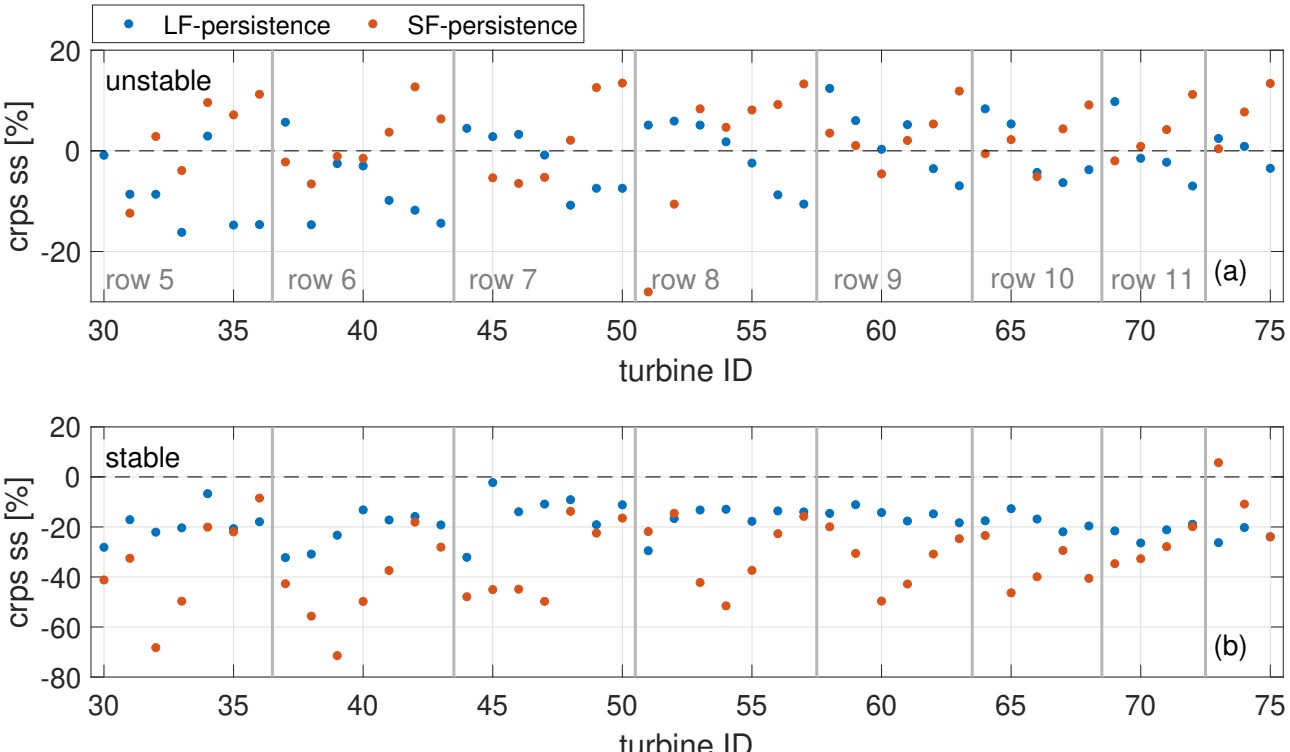

**Figure 4.** The crps ss of LF and SF with persistence as reference for individual turbines of GT I and distinguishing between (a) unstable and (b) stable atmospheric conditions. Grey vertical lines mark horizontal wind turbine rows.

fluctuates closely around zero for westerly winds. For southerly winds, scores are generally slightly larger and the bias of most turbines lies between 0.5 % and 1.5 %.

    The LF's dependency on lidar coverage was already shown in previous work (Theuer et al., 2020b). Here, we focused on the SF's sensitivity to missing turbine data. In case of failing measurement devices or maintenance operations, wind speed and wind direction information might be missing or inaccurate for some turbines during periods of time. Here, we analyzed

how the SF's forecast skill is affected by missing turbines. To do so, we randomly excluded an increasing amount of wind turbines as the origin of wind vector propagation for the whole analyzed time period. We will refer to the number of turbines considered as turbine availability in the following. In Figure 6 we compare the forecast availability and the $\overline{\text{crps}}$ normalized with respect to 100 % turbine availability for a number of exemplary turbines that have shown high forecast availability. The normalized $\overline{\text{crps}}$ in Figure 6 (b) only considers simultaneously available forecasts for all filter criteria. A reduction of turbine

availability clearly causes a decrease in forecast availability and skill for all of the analyzed turbines. The impact of missing turbines increases with lower turbine availability. For GT36, for instance, a reduction of turbine availability from 100 % to 50 % reduces the forecast availability to 97 % and increases the $\overline{\text{crps}}$ by 4.8 %. Further reducing turbine availability to only



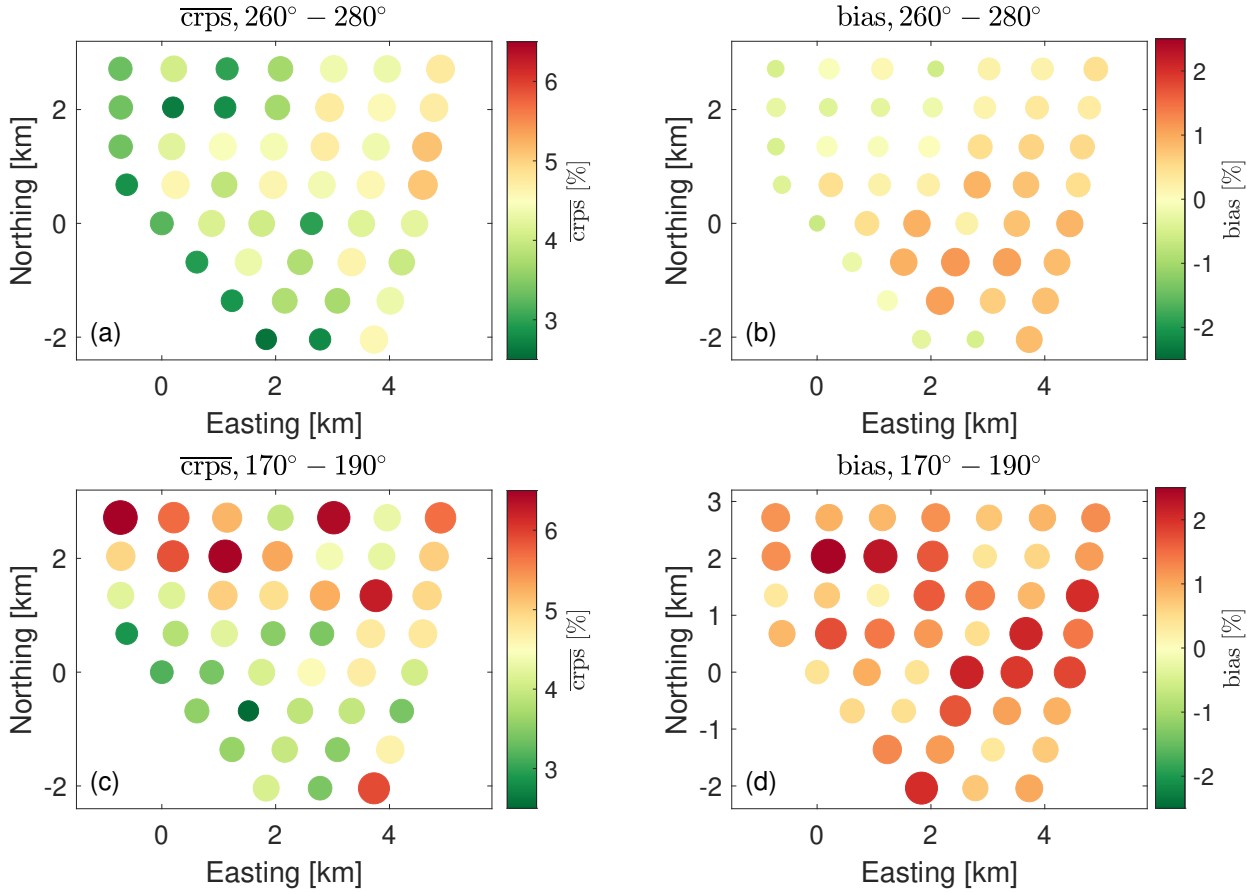

**Figure 5.** The $\overline{\mathrm{crps}}$ and bias of the LF for turbines GT30-GT75 in % of rated power for stable atmospheric conditions, situations below rated power and wind directions (a,b) $260° - 280°$ and (c,d) $170° - 190°$. The colour scale and magnitude of the dots visualize the magnitude of the scores.

25 %, lowers the forecast availability by another 10.6 % and increases the $\overline{\mathrm{crps}}$ by 11.5 %. A similar behaviour can be observed for turbines GT35 and GT42. Only for turbine GT56 the forecast availability and $\overline{\mathrm{crps}}$ change rather linear.

**3.3 Extension to an observer-based power forecast of individual wind turbines**

A main advantage of the OF compared to the LF or SF is its increased forecast availability. This is visualized in Figure 3, where the number of available forecasts for the 80 turbines of GT I for LF, SF and OF is shown. It becomes clear that the LF and SF complement each other well in terms of data availability (cf. Section 3.2) from which the OF can benefit. It shows high availability in the wind farm's centre, which decreases when approaching the north-westerly and south-easterly

region of the wind farm. This is a consequence of lidar trajectories, wind farm layout and wind conditions at the site. The




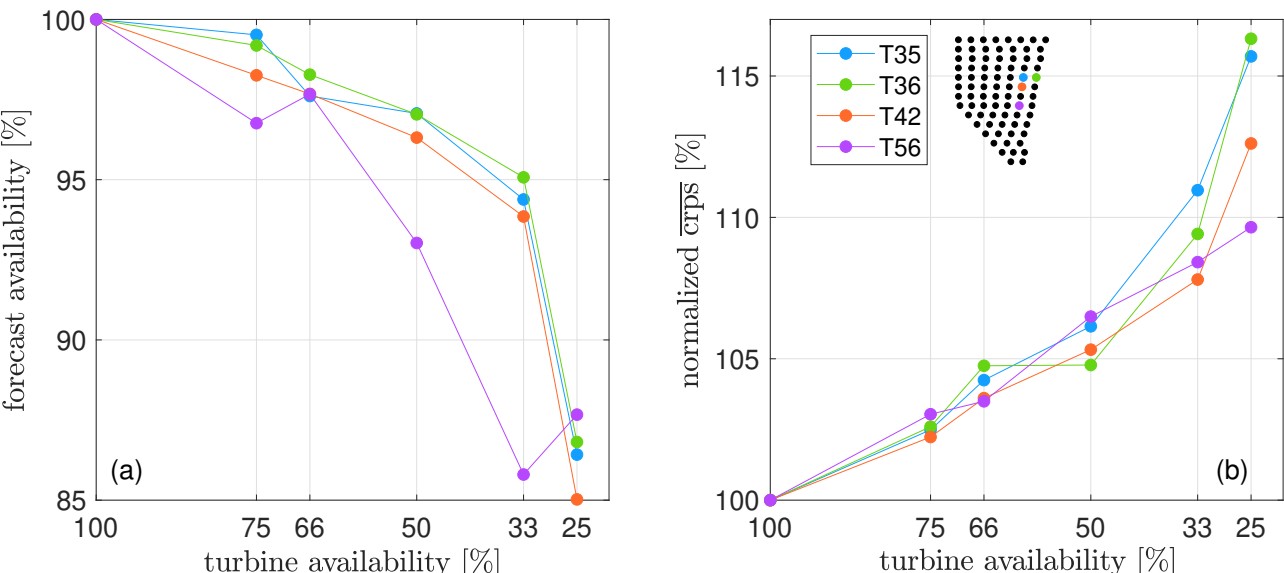

**Figure 6.** (a) Forecast availability in % and (b) $\overline{crps}$ normalized with respect to 100 % turbine availability in % for reduced turbine availability and selected example turbines. The wind farm layout visualizes the turbines' positions.

OF's availability for the selected turbines GT30-GT75 after filtering turbines during non-normal operation (cf. Section 3.2) is depicted in Figure 3 (d).

In addition to the forecast availability also the forecast skill can benefit from a combination of the two forecasting methodologies. Figure 7 depicts the $\overline{crps}$ for the OF compared to the LF, the SF and persistence for the 46 remaining turbines. To
be able to compare OF and LF respectively SF we only consider situations for which both of the forecasts are available. That means, in Figure 7 (a) we only take those OFs into account that consist of either a combination of LF and SF or solely the LF. We distinguish between unstable atmospheric conditions ($L < 0$) in blue and stable ones ($L > 0$) in red. The dot size represents the number of available forecasts at the respective turbine and is scaled with the maximal value of available forecasts within each subplot. Data positioned below the black diagonal line indicates an improvement of the OF's forecast skill compared to
the reference method.

In addition, in Figure 8 we present the crps skill score for the individual wind turbines distinguishing between atmospheric conditions for the same cases as visualized in Figure 7. The OF shows higher forecast skill for all turbines in both stable and unstable situations compared to the LF. It benefits strongest from additional SFs for turbines located far away from the lidar scans, which are most affected by the LF's long wind vector travelling distances and times and possibly by wake effects. A
number of turbines for which the effect almost disappears (e. g. GT44, GT51, GT58) indicated by dots positioned close to the diagonal line and a crps ss close to 0, are visible. Those correspond to free-stream turbines for which the amount of valid SFs is small and the OF consists mainly of LFs. Also compared to the SF, the OF's $\overline{crps}$ is improved for almost all analyzed turbines. The effect is most distinct during stable atmospheric conditions and for turbines close to the free-stream region of the



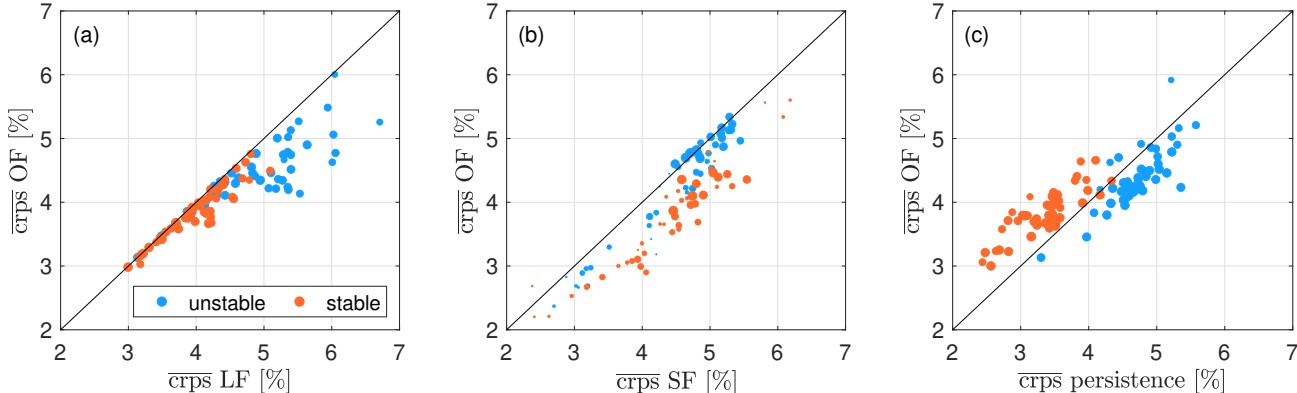

**Figure 7.** Comparison of $\overline{\mathrm{crps}}$ of the observer-based forecast to the (a) lidar-based forecast, (b) SCADA-based forecast and (c) persistence in % of the turbines' rated power. Each dot represents $\overline{\mathrm{crps}}$ for one of GT I's wind turbines (GT30-GT75) both for stable and unstable atmospheric conditions. The dot size scales with the number of forecasts considered. Only situations with forecasts available for both methods are considered.

wind farm (e. g. GT39, GT54, GT60), thus with few upstream turbines for the SF available. Here, the SF can benefit strongly

from additionally available lidar data.The OF is able to outperform persistence during unstable stratification for most turbines, however, it fails to do so during stable cases. Turbines for which the OF underperforms during unstable cases are positioned in the northerly region of the wind farm. Those located in the centre of the wind farm (e. g. GT50-GT58) can be forecasted best due to the beneficial data basis.

### 3.4 Calibration of observer-based power forecasts of individual wind turbines

Forecast calibration aims to improve the probabilistic characteristics of forecasts. Moreover, well-calibrated forecasts are a prerequisite for the application of the copula approach (cf. Section 2.4). In Figure 9 (a) we therefore compare the $\overline{\mathrm{crps}}$ of the raw and calibrated observer-based power forecast. As in Figure 7, we distinguish between atmospheric conditions and scale the marker size according to data availability. For almost all of the analyzed turbines the OF's skill was considerably improved by calibration. The effect seems most distinct for turbines with less accurate forecasts, which often coincide with lower data

availability. A comparison of the OF and persistence in Figure 9 (b) reveals that persistence is outperformed only for few of the turbines during stable atmospheric conditions. However, the OF is now more skillful than persistence during unstable situations for all analyzed turbines.

In addition to $\overline{\mathrm{crps}}$ we use reliability diagrams to evaluate the consistency between the statistics of the forecast and the observation. The reliability diagrams in Figure 10 visualize the analyzed quantile steps $[0, 0.1, ..., 1]$ on the x-axis. For each

time step the likelihood that a certain threshold is exceeded is determined from the forecast members and assigned to its specific quantile bin. The fraction of observations actually exceeding the threshold for those time steps is shown on the y-axis. In this case, we define a threshold of $0.9\,P_\mathrm{r}$. Accurate probabilistic forecasts of high power regimes are particularly important for grid integration and trading. The 95 % confidence intervals of the reliability diagrams are determined by means

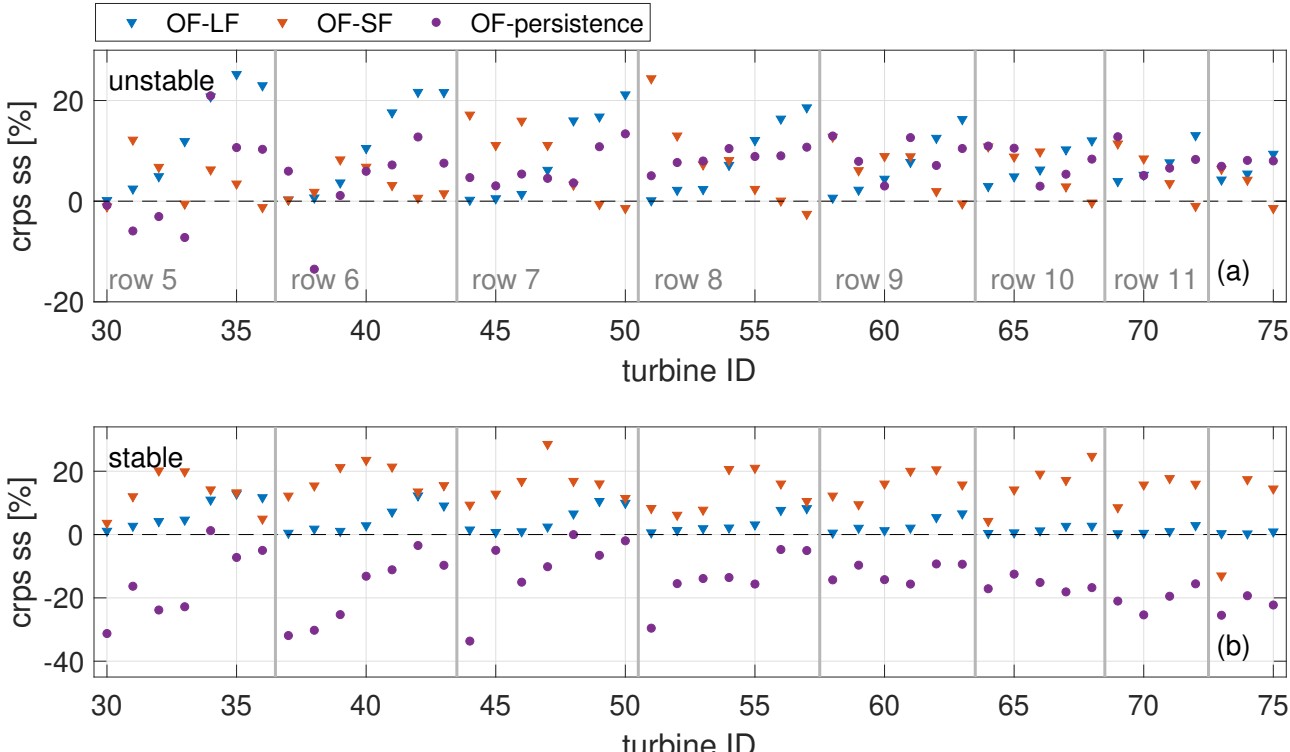

**Figure 8.** The crps ss of the OF with LF, SF and persistence as reference for individual turbines of GT I and distinguishing between (a) unstable and (b) stable atmospheric conditions. Grey vertical lines mark horizontal wind turbine rows.

of a bootstrapping approach and visualized as error bars. Due to the limited number of available forecasts, we did not distinguish
between atmospheric stability when evaluating reliability diagrams.

To analyze differences in reliability dependent on turbine location we selected the exemplary turbines GT30, GT57 and GT64. The reliability diagram of GT30 fluctuates more strongly around the diagonal and its confidence intervals are broad compared to GT57 and GT64. As visible in the histogram, this is related to a smaller number of valid forecasts, which in turn is a consequence of the turbine's location in the northerly region of the wind farm. In general, the data basis is too poor to draw
any conclusions from comparing the different methods or turbine locations. Overall, the OF seems reasonably well-calibrated.

### 3.5 Evaluation of aggregated wind turbine power forecasts

As explained in Section 3.1, the aggregation of individual turbines' power forecasts requires a large number of simultaneously available turbine forecasts. Furthermore, these individual forecasts need to be well-calibrated (Bessa, 2016). To have sufficiently large data sets that also allow for a distinction between atmospheric stability available we therefore limited our analysis
to a maximum number of seven turbines per subset. Turbines within one subset were selected as those in close proximity to each other to increase the number of simultaneously available forecasts. To test the copula approach for a number of differ-



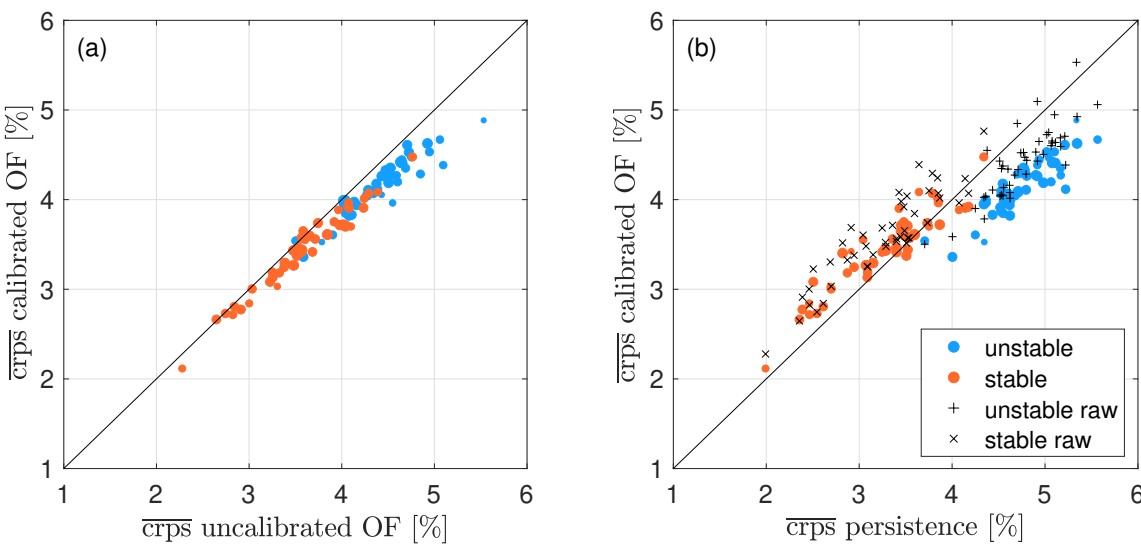

**Figure 9.** Scatterplots as described in Figure 7. In (a) calibrated and raw observer-based forecasts are compared and in (b) the calibrated observer-based forecast in color and the raw observer-based forecasts as black markers are compared to persistence.

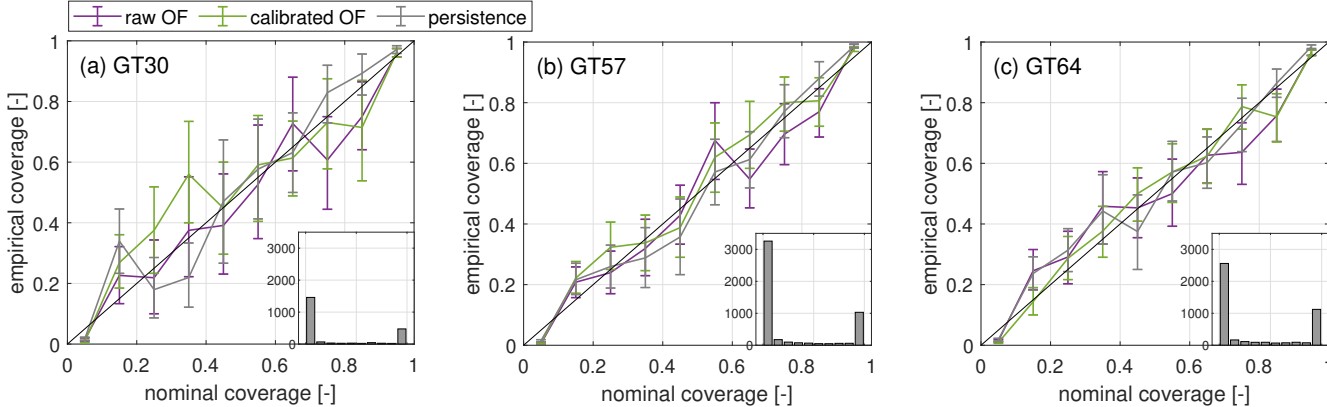

**Figure 10.** Reliability diagrams of the raw observer-based forecast (purple), the calibrated observer-based forecast (green) and persistence (grey) for turbines GT30, GT57 and GT64. 95 % confidence intervals are visualized as error bars. The black diagonal line indicates perfect reliability. Histograms show the number of valid forecasts per quantile step for the calibrated forecast.

ent circumstances, we selected subsets covering different parts of the wind farm, e. g. the westerly part in subset 1 and the easterly part in subset 3, and arranged in different shapes, e. g. an elongated turbine cluster stretching from the wind farm's south-westerly to north-easterly region in subset 2, a more dense cluster of turbines near the free-flow region in subset 4 or a
horizontal wind turbine row in subset 5.





In addition to probabilistic forecasts of aggregated wind turbine power, we also evaluated deterministic power forecasts using the root-mean-squared error (rmse)

$$\text{rmse} = \sqrt{\frac{1}{N}\sum_{i=1}^{N}(\text{fc}_i - \text{obs}_i)^2} \tag{16}$$

with forecasts $\text{fc}_i$ and observations $\text{obs}_i$ with time index $i$ and number of analyzed forecasts $N$.

We generated deterministic forecasts of turbine subsets by aggregating deterministic forecasts of individual turbines and refer to this method as deterministic OF in the following. Deterministic forecasts of individual turbines were determined by averaging their ensemble members. Additionally, the ensemble members of the subsets' probabilistic power forecasts determined using the three different copula approaches, namely the empirical Gaussian copula, the parametric Gaussian copula and the vine copula (cf. Section 2.4), were averaged. The turbine subsets used, the number of valid forecasts considered within each subset

and the results for the different copula approaches and persistence are summarized in Table 1 and 2 for unstable as well as stable atmospheric conditions. Further, reliability diagrams of all subsets and approaches are shown in Figure 11. The average absolute difference between empirical and nominal coverage for quantile steps $q$ and their number $N_q$ is summarized as quantile mean absolute error (mae)

$$\text{quantile mae} = \frac{1}{N_q}\sum_{q=1}^{N_q}|\text{empirical coverage}_q - \text{nominal coverage}_q|. \tag{17}$$

and additionally shown in Figure 11 (f).

In terms of $\overline{\text{crps}}$, 4 out of 5 subsets are able to outperform the benchmark persistence during unstable atmospheric conditions. For stable atmospheric conditions, persistence performs best. Generally, forecast skill is higher for the aggregated forecasts compared to those of individual turbines due to smoothing of power fluctuation averaging. For three subsets unstable atmospheric conditions can be predicted more accurately than stable situations by all evaluated methods, contradicting previous

results. A comparison of the different approaches and subsets with regard to their reliability and quantile mae is not conclusive, considering the overlap of the wide confidence intervals. This is a consequence of the small number of available forecasts. In terms of rmse, the copula approaches are able to outperform persistence for three and the deterministic OF for only one of the evaluated subsets during unstable atmospheric conditions (cf. Table 2). During stable cases, persistence is most accurate for all five subsets. Overall, scores are very similar for the three tested approaches and none of them can be identified as superior.

The analysis of the covariance matrices revealed their dynamic behaviour over time. The sliding-window approach allows the covariances to adapt to changing atmospheric conditions. In Figure 12 we show average empirical and exponential covariance matrices of subset 1 for different conditions. We distinguish between atmospheric stability, average power production of free-flow wind turbines (GT30, GT37, GT44, GT51, GT58, GT64, GT69, GT73) and average wind direction of turbines GT30-GT75. We select covariances considering conditions during the 6 h time window used for copula training. A comparison

of empirical (left, Figure 12 (a,c,e,g,i,k)) and exponential covariance matrices (right, Figure 12 (b,d,f,h,j,l)) makes clear that covariances are smoothed by the parameterization. For exponential covariances, a distinct dependency on the turbines' spacing





**Table 1.** Turbine subsets, number of valid forecasts considered and $\overline{\text{crps}}$ in % of the subsets' normalized power for the vine, empirical and exponential copula approach and persistence for unstable and stable atmospheric conditions. Lowest scores are shown in bold.

|  | subset | 1 | 2 | 3 | 4 | 5 |
|---|---|---|---|---|---|---|
|  | turbines | 45, 46, 52, 58, 59, 65 | 40, 45, 46, 52, 58 | 42, 43, 48, 50, 55, 56, 57 | 51, 52, 58, 59, 64, 65 | 51, 52, 53, 54, 55, 56, 57 |
|  | $N$ | 1012 | 1101 | 612 | 1074 | 876 |
| $\overline{\text{crps}}$ [%] unstable | persistence | 2.32 | 2.99 | 3.39 | **2.29** | 2.20 |
|  | vine | **2.20** | 2.68 | **2.82** | 2.42 | **2.05** |
|  | empirical | 2.21 | 2.68 | 2.83 | 2.42 | 2.07 |
|  | exponential | 2.21 | **2.67** | 2.85 | 2.42 | 2.08 |
|  | $N$ | 529 | 489 | 279 | 537 | 350 |
| $\overline{\text{crps}}$ [%] stable | persistence | **2.88** | **2.31** | **2.89** | **2.57** | **2.81** |
|  | vine | 2.94 | 2.61 | 3.14 | 2.80 | 3.04 |
|  | empirical | 2.94 | 2.59 | 3.12 | 2.80 | 3.03 |
|  | exponential | 2.95 | 2.59 | 3.12 | 2.80 | 3.02 |

**Table 2.** The rmse in % of the subsets' normalized power for the vine, empirical and exponential copula approaches, the deterministic approach and persistence for unstable and stable atmospheric conditions. Lowest scores are shown in bold.

|  | subset | 1 | 2 | 3 | 4 | 5 |
|---|---|---|---|---|---|---|
| **rmse** [%] unstable | persistence | 5.17 | 6.12 | 6.32 | **4.59** | **4.80** |
|  | deterministic | **5.08** | **5.59** | 5.58 | 5.07 | 4.92 |
|  | vine | 5.11 | 5.65 | 5.56 | 5.09 | 4.92 |
|  | empirical | 5.11 | 5.67 | **5.54** | 5.10 | 4.94 |
|  | exponential | 5.11 | 5.66 | 5.58 | 5.09 | 4.94 |
| **rmse** [%] stable | persistence | **5.25** | 4.21 | **5.41** | **4.51** | **5.01** |
|  | deterministic | 5.59 | 5.02 | 5.81 | 5.24 | 5.58 |
|  | vine | 5.61 | 5.03 | 5.84 | 5.27 | 5.59 |
|  | empirical | 5.62 | **5.01** | 5.86 | 5.29 | 5.61 |
|  | exponential | 5.62 | **5.01** | 5.82 | 5.28 | 5.59 |

can be observed. Figure 12 (a)-(d) show that, as expected, covariances are on average higher during stable atmospheric conditions than during unstable cases. In Figure 12 (e)-(h) we compare covariances of situations with turbines operating below rated power ($< 0.9\,P_{\text{r}}$) and those running at rated power ($\geq 0.9\,P_{\text{r}}$). Slightly larger values can be observed below rated power.

In Figure 12 (i)-(l) we analyze the covariances' dependency on wind direction. To exclude the impact of atmospheric stability and power production, we only consider cases with stable stratification and turbines operating below rated power here. To maximize the number of valid covariance matrices, wind direction intervals are chosen relatively large with $240° - 300°$ and $< 240°$. Overall, covariances are higher for westerly winds as compared to south and south-westerly winds. We relate this mainly to changing wake situations. We exemplarily analyze the covariances' dependency on wind direction using turbine

pairs GT45-GT46, GT45-GT52 and GT46-GT52. While for westerly winds the average covariance of GT45-GT52 is higher

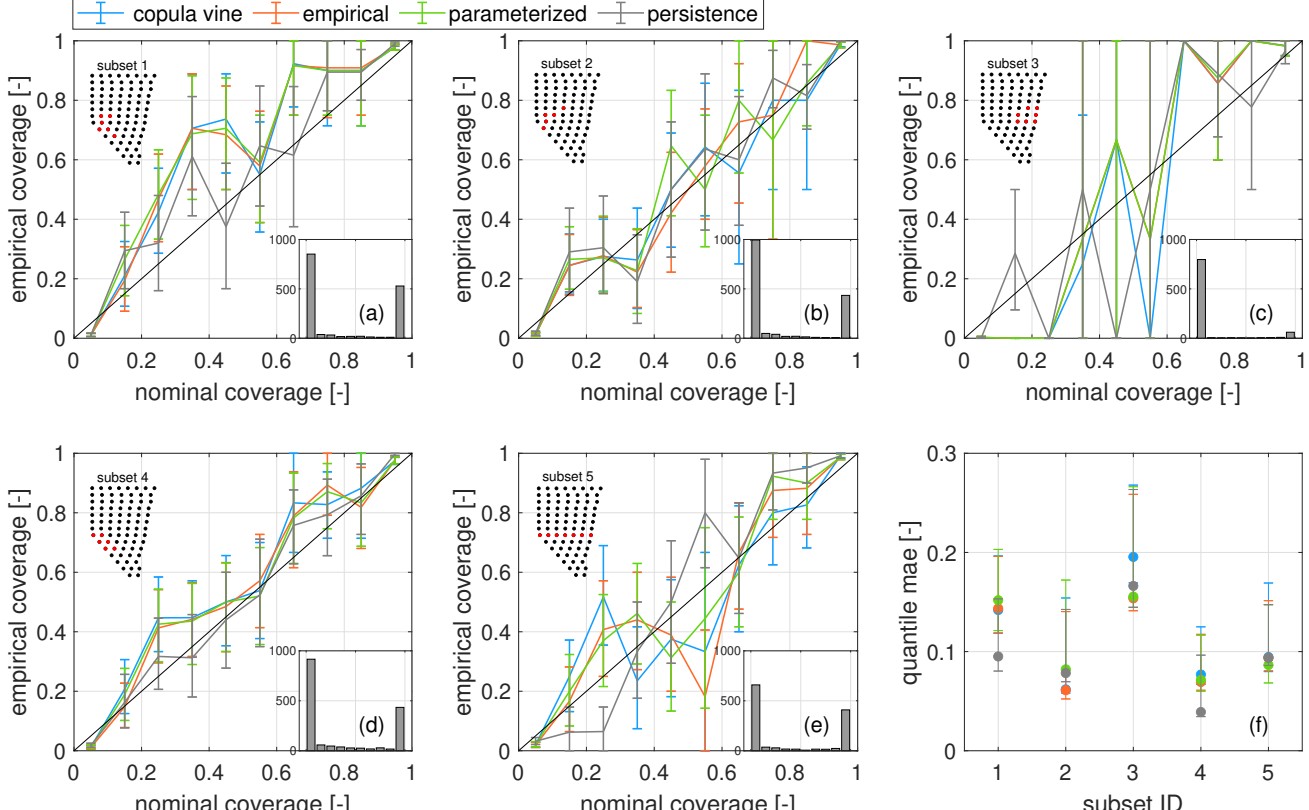

**Figure 11.** (a)-(e) Reliability diagrams for the different turbine subsets (1-5) and copula approaches summarized in Table 1 and persistence. The turbine subsets are marked in red in the small wind farm layouts. Histograms show the number of valid forecasts per quantile step for the empirical copula approach. The black diagonal line indicates perfect reliability. In (f) the corresponding quantile maes are shown. For all subfigures 95 % confidence intervals are visualized as error bars.

than that of GT45-GT46 and GT45-GT52, it is lower for south and south-westerly winds. This can be explained because for westerly winds, GT45 and GT52 experience similar wake conditions and are positioned approximately perpendicular to the incoming wind. In contrast, for south and south-westerly winds, their wake situation is different, with GT52 placed upstream of GT45. Here, GT45-GT46 and GT46-GT52 are subject to more similar wake effects and exhibit higher covariances. It should

be noted that the number of covariance matrices considered for the different filter criteria varies considerably.

## 4   Discussion

In the following, we review the lidar- and SCADA-based forecasting methodologies with regard to the impact of wakes and data availability. Further, the generation and calibration of the observer-based forecast as well as the aggregation of individual power forecasts by means of a copula approach is discussed. Finally, we assess the value of minute-scale power forecasts of

offshore wind in a broader context.



**Figure 12.** Average covariances of turbine subset 1 determined using the empirical (left: a,c,e,g,i,k) and exponential (right: b,d,f,h,j,l) copula approaches. Unstable (a,b) and stable (c,d) cases and situations with turbines operating below ($< 0.9P_r$) (e,f) and at rated power ($\geq 0.9P_r$) (g,h) are compared. In (i),(j),(k) and (l) stable situations with turbines operating below rated power with wind directions ranging from $240° - 300°$ (i,j) and $< 240°$ (k,l) are depicted. The turbine subset 1 is marked in red in the small wind farm layout.

## 4.1 Lidar- and SCADA-based power forecasts of individual wind turbines

In previous work (Theuer et al., 2020b, 2021) we have focused on the forecast of the first row of wind turbines, with respect to the main wind direction, only. Here, we extended the forecast to all wind turbines of the wind farm, also including waked wind turbines. Generally, the LF's skill is highest for free flow turbines and areas covered well by the lidar scans. Our analysis

has revealed that forecasting errors are larger for wind turbines and wind directions directly impacted by wakes, while a systematic over- or underestimation of wind speed was not observed. That means, the LF is generally able to capture the mean wake effect, however, not able to forecast small scale fluctuations associated with it. The LF considers, just like persistence, past observations at the turbine of interest that are then multiplied with the wind speed tendency determined from lidar data (cf. Section 2.1). It is thus able to account for wakes to some extent. We assume that the higher errors observed are mainly



related to turbulence in wake regions that cannot be represented well by Lagrangian advection. Furthermore, wind vectors reaching turbines positioned in the easterly and north-easterly region of the wind farm were typically propagated over a longer distance and time compared to turbines closer to the lidar scans. These vectors can be associated with higher uncertainty. For the SF, forecasts are most accurate in the region of the wind farm opposite to the prevailing wind direction, i. e. the north-easterly region. Here, the applied bias-correction prevents systematic errors. Wind vector propagation of the SF is affected

more strongly by wakes than the LF as it is performed at hub height. Also Valldecabres et al. (2020) accounted for wakes in their work by applying a directional turbine efficiency, which significantly improved their results. However, the forecast was only able to outperform persistence in terms of $\overline{crps}$ for wake-influenced turbines during ramp events.

The SCADA-based forecast introduced in this work is based on a high-frequency (0.2 Hz) flow reconstruction and prediction methodology developed by Rott et al. (2020). We extended this work to a probabilistic approach by resampling the selected

wind vectors also considering the weights assigned to them and included a power transformation. Rott et al. (2020) applied and validated their model to a high-frequency data set, aiming at applications in wind turbine control. In our work, we focus on 1-minute-mean forecasts with a temporal resolution of 2.5 minutes, in accordance with the lidar scans. Therefore, we adjusted the methodology to preselect wind vectors following the lidar-based forecasting methodology, considering only those reaching an area of influence within a certain time window before applying the inverse temporal distance weighting. As opposed to

Rott et al. (2020), we neglected the spatial distance weighting and relied solely on the temporal distance weighting, using a Shepard parameter of $p = 4$. Rott et al. (2020) state that the usage of large Shepard parameters results in a more accurate representation of wind speed fluctuations, while lower parameters allow a robust forecast of average wind speeds. We chose a medium parameter as a good compromise between robustness and temporal resolution of wind speed fluctuations.

While the flow reconstruction method was applied only to forecasts with lead times up to 120 s, the results indicated that an

application to forecasts with larger lead times might be valuable. Rott et al. (2020) showed that forecast accuracy decreases with lead time, however, its skill compared to persistence increases. Our results confirm the methodology's benefit compared to persistence for lead times of 5 minutes. Inaccurate wind direction data might impact the accuracy of SCADA-based forecasts. Wind direction was determined using the absolute yaw position and wind vane of each turbine, both of which are subject to uncertainties (Mittelmeier and Kühn, 2018; Simley et al., 2021). Rott et al. (2020) identified the model's approach to consider

wakes and disturbances of the sonic anemometers and consequently wind direction measurements as additional sources of uncertainty.

The SF is able to account for missing data to some extent. It can thus be considered robust against lacking data of individual wind turbines that might occur during daily operation of a wind farm due to maintenance or failing measurement devices. Only with more distinct reductions of turbine availability forecast skill and forecast availability were significantly reduced. In that

case, gaps are too large and important information is lost. How strongly missing turbine data impacts forecast accuracy is also dependent on wind speed, wind direction and the target turbine's position. They could, just like insufficient lidar coverage, cause systematic forecasting errors.



## 4.2 Extension to an observer-based power forecast, forecast calibration and aggregation

The lidar- and SCADA-based forecasts complement each other well in terms of data availability. Further, the forecast skill of
the observer-based forecast outperforms both individual methods. Our analysis clearly showed that both forecasting methods, LF and SF, profit from the additional data set considered in the OF. While we relate this mainly to an improved data basis for certain areas of the wind farm, a combination can also benefit from the individual forecasts' methodical differences. During unstable situations the SF was most significantly improved for turbines close to free-flow turbines due to significantly improved coverage. For stable stratification, the largest improvement shifts to turbines located further downstream. We relate this to
more pronounced wake effects during stable stratification. As suggested previously, the LF is able to account for wakes more accurately than the SF (cf. Section 3.2 and Section 4.1), which means it can significantly increase the SF's value in such situations. For turbines located far away from the lidar, when propagated lidar wind vectors are associated with high uncertainty due to wakes and their increased propagation distance and time, the OF mainly benefits from more recent SCADA wind vectors.

It is common practice in (power) forecasting to combine different forecasting approaches to improve performance. Junk et al.
(2015), for instance, combined different Ensemble Prediction Systems to multi-model ensembles. They introduced different weighting approaches, namely implicit weighting, equal weighting and optimized weighting. The authors found that optimized weighting did not improve forecast calibration, while implicit weighting, which is based on the different number of ensemble members of the models, performed best. In our work, we were not able to apply implicit weighting as the number of wind vectors selected for the forecast strongly depends on the different spatial and temporal scales of the data sources. Future work
should analyze how the different numbers of wind vectors reaching a certain turbine using the LF or SF can be considered in the weighting, thus moving from the equal-weighting approach to a more implicit one.

Forecast calibration by means of Ensemble Model Output Statistics allows to correct for systematic errors as well as ensemble spread. By using a moving-time-window approach it is also possible to account for systematic errors varying with atmospheric conditions, for instance wind direction-dependent wake losses. Varying atmospheric stability and turbulence in-
tensity that might impact power fluctuations can be addressed by adapting the forecast spread.

As we were only able to aggregate a maximum of seven turbines, it is not yet possible to draw any conclusion regarding the copula approach's ability to predict the total wind farm power. Results indicate, however, that copulas can be a valuable tool to support the generation of probabilistic forecasts. Even though we generally expect persistence to have an advantage compared to observer-based methods for aggregated wind power forecasts as power fluctuations are averaged out, persistence
underperformed for four out of five subsets in terms of $\overline{crps}$ during unstable conditions. The higher skill during unstable situations compared to stable ones for three of the analyzed subsets contradicts previous results (Theuer et al., 2020b, 2021). It is likely related to a higher number of situations with turbines operating at rated power, which are associated with a higher forecast skill. Gilbert et al. (2020) applied a similar methodology to aggregate individual wind turbines' power forecasts and were also able to beat two benchmarks, namely a quantile regression model and an Analog Ensemble method. However,
their forecast's lead time was much larger, its temporal resolution much lower and a distinction between stability cases was not made, making a comparison difficult. The high temporal resolution of the OF might be one reason why covariances in



our study are generally lower compared to the results of Gilbert et al. (2020). We found the magnitude of covariances to be dependent on atmospheric stability, turbine spacing, power production and wind direction. The small data set makes a more detailed distinction between different conditions difficult. Covariances are lower in situations with many power fluctuations, as expected during unstable atmospheric conditions and when turbines are subjected to wakes. Also for high power regimes, when typically the ensemble spread is narrow, quantiles are less correlated and thus the covariances are low. In cases where power forecasts and actual power production of neighbouring turbines can be expected to be rather similar, covariances are higher. This might happen due to more homogeneous wind fields upstream, typically during stable atmospheric conditions and when the impact of wakes on the neighbouring turbines is similar.

An analysis of the rmse revealed that for deterministic forecasts of turbine subsets it is more skillful to aggregate individual deterministic wind turbine forecasts. The comparison of different copula approaches suggests the use of an empirical or parametric copula instead of a vine copula. Vine copulas are more computationally expensive, however, able to achieve only marginally better results. Similar conclusions were drawn by Bessa (2016) and Gilbert et al. (2020). Results also varied for different turbine subsets. This is possibly related to different numbers of turbines considered, the different skill of the individual turbines' forecasts or varying distributions of atmospheric conditions within the data sets. For Sklar's theorem to hold, marginal distributions of forecasts need to be uniformly distributed. While our forecasts were reasonably well calibrated, further improvement would possibly also have benefits in the copula generation.

### 4.3 Future value of minute-scale offshore wind power forecasts

For future minute-scale forecasts of offshore wind power, considering, for example, the large number of wind farms in the North Sea and also their close proximity to each other, it might be beneficial to include operational data of several wind farms into the observer-based forecast. We expect that these additional data sources could further increase data availability, enhance forecast skill and in particular enlarge the forecast horizon. In such a case, however, one would need to carefully calibrate the forecast to include operational data from different wind farms. The availability of lidar-based forecasts could further be increased by deploying several lidar devices and by developing more powerful lidars, e. g. with considerably increased range or scanning speed. This might facilitate multi-elevation scans with a better resolution of the rotor swept area of future very large offshore turbines.

The forecast skill of lidar-based, SCADA-based and consequently observer-based forecasts is expected to decrease with increasing lead time as a consequence of assumptions made during Lagrangian advection as discussed in previous studies (Würth et al., 2018; Rott et al., 2020; Theuer et al., 2020b). An observer-based forecast covering large areas of e. g. the North Sea is therefore not expected to be able to forecast small scale structures very accurately. However, it would likely be able to predict the occurrence of power ramps caused, for example, by passing fronts. It was shown in numerous studies and confirmed in this work that remote sensing-based forecasts are able to outperform persistence in particular during unstable or turbulent situations and also during ramp events (Valldecabres et al., 2020; Theuer et al., 2021). The development of an early-warning-system of potentially grid-critical power ramps based on observer-based forecasts covering the North Sea is therefore considered a valuable extension to persistence.



The overall value of observer-based forecasts compared to persistence for longer time periods will strongly depend on typical atmospheric conditions at the wind farm site. During stable atmospheric conditions forecasts are generally more accurate, but the OF is not able to outperform persistence (Theuer et al., 2021). In those cases, it should be considered applying persistence instead or possibly a hybrid model that includes persistence (Theuer et al., 2022).

The aggregation of individual wind turbine power forecasts using a copula approach was strongly restricted by limited data availability in this work. As shown in other work (Valldecabres et al., 2018a; Theuer et al., 2020b) and previously discussed the availability of forecasts is strongly dependent on lidar trajectories, wind farm layout and wind conditions. Excluding certain operating conditions of turbines further reduced the available data set. That means, in particular for a wind farm as large as Global Tech I, the generation of reliable simultaneously available forecasts for all turbines is difficult. Further analysis is

required to evaluate how the proposed methods might benefit probabilistic power forecasts for wind farms of smaller size or with an overall higher forecast availability. Also trajectory optimization or the installation of multiple lidars instead of just one could improve the applicability of the copula approach. To evaluate the benefit of hierarchical forecasting these methods should also be compared to wind farm power forecasts that do not consider individual power forecasts on the turbine level (Pichault et al., 2021).

**5   Conclusions**

We developed an observer-based minute-scale offshore wind power forecast by combining a lidar-based and a SCADA-based approach. To improve probabilistic forecast skill we calibrated the observer-based approach. Further, a copula methodology was implemented to generate probabilistic power forecasts of aggregated turbine subsets.

    Our results revealed the high potential of a complementary use of lidar-based and SCADA-based forecasts regarding both

forecast availability and skill. We conclude that a combination of SCADA- and lidar-based forecasts is beneficial for all turbines in the wind farm and during both stable and unstable atmospheric conditions. Lidar-based forecasts were less skillful for wake-influenced turbines than for free-stream ones, however, able to predict the mean wake effect. SCADA-based forecasts were found to be very robust against reduced turbine availability. To guarantee high availability and skill of lidar-based forecasts a careful planning of lidar scanning trajectories is required, considering main wind direction, wind farm layout and lidar

capabilities.

    Forecast calibration was found to significantly reduce the forecasts' average crps, however, as a consequence of the small data set no conlcusions regarding the calibration's impact on reliability could be drawn. Even though forecast skill was significantly improved compared to the raw forecasts, calibrated observer-based forecasts were only able to outperform persistence during unstable rather than stable atmospheric conditions. Based on these results we conclude that for an operational use of the

observer-based forecast a distinction between atmospheric conditions is useful. Given the current status of the methodology, during stable conditions it is recommended to rely on persistence. Also the use of a hybrid methodology might be beneficial and should be explored in the future. Applying the copula approach to generate aggregated probabilistic power forecasts for turbine





subsets showed high potential. Empirical and parametric covariance matrices were found advantageous over vine copulas in particular considering their high computational cost. The copula approach was not able to add value to deterministic forecasts.

In future work the copula approach for probabilistic minute-scale power forecasting needs to be further analyzed for wind farms with higher overall forecast availability.

*Data availability.*   Lidar and meteorological data are not published and could be made available on request. The OSTIA data set can be obtained from http://marine.copernicus.eu (Copernicus marine service, 2022). GT I SCADA data is confidential and therefore not available to the public.

*Author contributions.*   FT conducted the main research and was lead author of the paper. JS conducted the measurement campaign, supported lidar data analysis, contributed to the scientific discussion and provided extensive feedback in form of manuscript reviews. AR supported the development of the observer-based forecast, gave extensive feedback on copula and calibration methods and their mathematical formulation and reviewed the manuscript. LvB and MK supervised the work, contributed to the scientific discussion, the structure of the paper and thoroughly reviewed the manuscript.

*Competing interests.*   The authors declare no conflict of interests.

*Acknowledgements.*   We thank the German Federal Environmental Foundation (DBU) as this project receives funding within the scope of their PhD scholarship program (Grant Nr. 20018/582). The lidar measurements and parts of the work were performed within the research projects OWP Control (Ref. Nr. 0324131A), WIMS-Cluster (Ref. Nr. 0324005) and WindRamp (Ref. Nr. 03EE3027A) funded by the German Federal Ministry for Economic Affairs and Climate Action on the basis of a decision by the German Bundestag. We acknowledge the wind
farm operator Global Tech I Offshore Wind GmbH for providing SCADA data and thank them for supporting the measurement campaign in GT I and our work. We acknowledge the UK Met Office for making the OSTIA data set available. We thank Stephan Stone for conducting the measurement campaign and Marijn Floris van Dooren for numerous scientific discussions on the forecasting methodology.



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
