# Peer review of "Observer-based power forecast of individual and aggregated offshore wind turbines"

_Wind Energy Science, 2022_

## Referee Comment (RC1)

**Review of the manuscript entitled "Observer-based power forecast of individual and aggregated offshore wind turbines," written by Frauke Theuer, Andreas Rott, Jörge Schneemann, Lueder von Breman, and Martin Kühn.**

In this manuscript, the authors propose a combination of remote sensing LiDAR forecasting techniques and SCADA-based techniques to predict individual and aggregate wind turbine performance. Calibration, as well as probabilistic predictions, are discussed, and the results are compared with benchmark prediction methods. In unstable atmospheric conditions, the proposed approach improves upon the benchmark cases.

The introduction of the manuscript provides clear motivation and relevant background as well as indicates how the current work will add to the current literature. Section 2 aims to introduce the forecasting methods used in the work. Previous LiDAR forecasting methods are first introduced, followed by SCADA methods. Finally, the combined prediction method is explained. All explanations are straightforward and build logically on each other so that the final forecast method for individual turbines is well understood. Finally, the individual probabilistic forecasts are aggregated into a farm-level probabilistic forecast. Overall, this section is good. The main comment is that it is not always clear which contributions are the authors' and which predated the manuscript.

The next section discusses the results of the forecasting methods in a case study. First, the experimental campaign is described as well as the forecasting parameters specific to the site. In the paragraph beginning at line 208, further justification could be given regarding the selection of these parameters. The forecasting approach is applied first to individual turbines. The strengths of the LiDAR-based approach - predicting turbines near incoming conditions - as well as the strengths of the SCADA-based approach - predicting inner turbines - are shown to complement well. Accuracy of forecasts with wind direction and considering cases where turbines may not be capturing data is discussed. Next, the two approaches are combined into the observer-based approach. The complementary nature of the LiDAR and SCADA approaches is clear in the increased forecast availability and accuracy of the observer approach. However, the persistence method is still shown to outperform the observer approach in stable conditions. Calibration is shown to improve observer performance. Finally, the aggregate forecasts are evaluated as well as deterministic forecasts. The benchmark persistence method was improved upon by observer models for unstable conditions but not stable conditions. Overall, this section is excellent, providing a logical evaluation of the models, a clear rationale for the model behavior, and good support for the use of observer methods in unstable conditions.

The final two sections first discuss the impact of the observer method given other forecasting methods and specific implementations and then conclude the manuscript. The discussion section is valuable in placing the observer method in the broader context of application in other sites while

considering its usefulness compared to other methods. The final section concludes the paper while highlighting areas for future improvement. These two sections combined do very well to highlight the potential of the observer method while also conceding that further improvements are possible.

Additional comments are listed below:

**Comments:**

- Section number is missing in reference on line 78.
- Paragraph not indented at line 208.
- In figure 3, keep all subplots the same size; it would be helpful to indicate on a subplot illustrating the entire farm (a-c) the region highlighted in the subset (d).

---

## Author Response (AR1)

**Author's response to the referee comments regarding the paper**

**Observer-based power forecast of individual and aggregated offshore turbines**

**Frauke Theuer, Andreas Rott, Jörge Schneemann, Lueder von Bremen, and Martin Kühn**

We thank the two referees for taking the time to read and review our manuscript and appreciate their feedback and comments. We address each of the remarks below. Please also find a pdf file with changes to the manuscript marked using latexdiff attached.

Anonymous Referee 1:

*The main comment is that it is not always clear which contributions are the authors' and which predated the manuscript. [referring to Section 2]*

Thanks for this comment. We added additionally references throughout the whole Section 2 to better distinguish between the new contributions of this manuscript, our own but predating contributions and other researchers work.

*In the paragraph beginning at line 208, further justification could be given regarding the selection of these parameters.*

We have adjusted the paragraph to make our choice of parameters more clear.

Each forecasted time step of the LF considered the six most recent scans, thus can contain wind data measured during the last 15 minutes. This ensures that also turbines positioned far away from the lidar scans can be reached by low wind speeds and their forecasts will not be biased. Wind vectors contributing to the SF were weighted using a tuning parameter of $p = 4$. The choice of this parameter is further discussed in Section 4.1. The SF's bias correction was performed considering a number of $N_\mathrm{t} = 5$ time steps prior to forecast initialization. This ensures that there is enough data for bias estimation while keeping the correlation high. The step length was chosen as $\Delta\tau = 156\,\mathrm{s}$ in accordance with that of the lidar scans. LF and SF were generated with an area of influence of $2D$ and a minimum of 20 required wind vectors (Theuer et al., 2021) and were resampled to contain 500 members. Forecast calibration was performed with a 5 h training interval before forecast initialization. The time window was optimized in a sensitivity analysis. A calibration was only performed for situations with at least 60 % valid data within that training period.

*Section number is missing in reference on line 78.*

We slightly changed the sentence to make the reference more clear.

In this work this approach is significantly extended further as described in the following: Using SCADA (Supervisory Control and Data Acquisition) data, it is first extended to an observer-based forecast (OF) to increase forecast availability and skill (cf. Section 2.2). In a next step, observer-based forecasts are calibrated by means of Ensemble Model Output Statistics (EMOS) (cf. Section 2.3). Finally, probabilistic power forecasts of individual wind turbines are aggregated using different copula approaches (cf. Section 2.4.)

*Paragraph not indented at line 208.*

We have corrected this.

40 *In figure 3, keep all subplots the same size; it would be helpful to indicate on a subplot illustrating the entire farm (a-c) the region highlighted in the subset (d).*

We have adjusted the size of Figure 3 (d). It now shows all turbines, with those not considered for further analysis in grey. The figure caption was adjusted accordingly.

45 **Anonymous Referee 2:**

*1. Line 92: Please describe briefly what is meant by "synchronized in time", such that the reader may follow your argument without having to study the paper referenced.*

We added a more detailed explanation of the time synchronization in Line 92f..

50 After wind field reconstruction, the individual lidar scans are interpolated to a cartesian grid and synchronized in time (Beck and Kühn, 2019). Time synchronization refers to the propagation of individual parts of the lidar scans measured at different times to the same time step using semi-Lagrangian advection. It aims at accounting for the large time shift within each scan.

*2. Figure 1: The Windpark "Hohe See", next to Global Tech I has been in operation since mid August 2019; your mea-*
55 *surement campaign ended in June 2019. Have you seen any influence of Hohe See in your data? In any case, it would be appropriate to indicate Hohe See in Figure 1a as well (e.g. using grey dots) and clarify the impact (or the lack thereof).*

We added the layouts of the wind farms Albatros (+) and Hohe See ($\times$) in Figure 1 and adjusted the caption accordingly. Before those wind farms started to operate we were occasionally able to observe the turbines' transition piece as hard targets in the lidar scans. Those hard targets were omitted during data filtering and thus did not impact the forecast. We added a
60 corresponding statement in Line 206f.:

Figure 1 (a) additionally depicts the layout of the wind farms Albatros and Hohe See, which were under construction during the time of the analysis. Those turbines were visible as hard targets in the lidar scans occasionally, which were omitted during data filtering an thus did not impact the forecast.

65 *3. Chapter 3.1 (199ff): This information is clearly important, but it should be presented in the Methods section, not as Results.*

We thought about including the information related to the specific campaign and data set in the Methods section, however, decided against it. In Section 2 we are aiming to introduce the methodology in a generic way, independently from the case study analysed in the manuscript. In our opinion it is therefore more suited to include the description of the case study in Section 3,
70 as we also did in our previous study Theuer et al. (2020). Other examples of journal papers using a similar structure are Bessa (2016), Pinson et al. (2009), Schuhen et al. (2012) and Gneiting et al. (2005).

*4. Line 245: I believe you want to refer to Figure 3, not 2.*

You're right, we changed the reference to Figure 3 (a).

75

*5. Table 12: "normalized power" means $P/P_r$, I assume? Please clarify.*

Thanks for pointing this out. This is a mistake and is meant to read *rated power*. We corrected it in the manuscript.

*6. Discussion: I was wondering if you could comment on the expected skill of the presented method for e.g. a 10 minute forecast. Is the range of the lidar the limiting factor, or the simple propagation technique?*

The skill of observer-based forecasts in general is expected to decrease with increasing forecast horizon as discussed in line 492ff. and numerous other studies (Würth et al., 2018; Rott et al., 2020; Theuer et al., 2020). Hereby, the range of the lidar (respectively the position of the turbines for the SCADA-based approach) will determine the maximal lead time for which it is possible to generate a forecast. At this stage not yet considering the skill of the forecast, here it is of concern if wind vectors are reaching the turbine of interest at the time of interest. This is also dependent on the wind farm layout, wind speed and direction and the position and scanning trajectory of the lidar (Theuer et al., 2020).

In addition to the availability of the forecast the lidar range will also impact the skill of a lidar-based forecast. Wind speed measurements from far ranges of the lidar scans are associated with larger uncertainty. Further, if wind vectors are primarily selected from the maximal measurement distance of the lidar, the forecast will more likely be biased. In our specific campaign these considerations restricted us to the evaluation of 5-minute ahead forecasts.

Apart from the lidar range, also the wind vector propagation has a large impact on forecast skill. The uncertainty associated with Lagrangian advection is expected to increase for larger propagation distances and duration, thus also with increasing forecast horizon. A first evaluation of lidar-based forecasts with forecast horizons up to 30 minutes based on a currently ongoing campaign confirms this. Further analysis will investigate, among other things, how the forecast skill for larger horizons compares to that of persistence. We expect that the lidar-based forecast is able to outperform persistence for horizons larger than 5 minutes, in particularly during very unstable atmospheric conditions and during ramp events.

As we already discuss the impact of forecast horizon on skill in Section 4.3, Line 492-504, in Theuer et al. (2020) and Theuer et al. (2021) in some detail we did not add further discussion in the manuscript.

**References**

100  Beck, H. and Kühn, M.: Temporal Up-Sampling of Planar Long-Range Doppler LiDAR Wind Speed Measurements Using Space-Time Conversion, Remote Sensing, 11, 867, https://doi.org/10.3390/rs11070867, 2019.

Bessa, R. J.: On the quality of the Gaussian copula for multi-temporal decision-making problems, in: 2016 Power Systems Computation Conference (PSCC), pp. 1–7, https://doi.org/10.1109/PSCC.2016.7541001, 2016.

Gneiting, T., Raftery, A. E., Westveld, A. H., and Goldman, T.: Calibrated Probabilistic Forecasting Using Ensemble Model Output Statistics
105  and Minimum CRPS Estimation, Monthly Weather Review, 133, 1098 – 1118, https://doi.org/10.1175/MWR2904.1, 2005.

Pinson, P., Madsen, H., Nielsen, H. A., Papaefthymiou, G., and Klöckl, B.: From probabilistic forecasts to statistical scenarios of short-term wind power production, Wind Energy, 12, 51–62, https://doi.org/10.1002/we.284, 2009.

Rott, A., Petrović, V., and Kühn, M.: Wind farm flow reconstruction and prediction from high frequency SCADA Data, Journal of Physics: Conference Series, 1618, 062 067, https://doi.org/10.1088/1742-6596/1618/6/062067, 2020.

110  Schuhen, N., Thorarinsdottir, T. L., and Gneiting, T.: Ensemble Model Output Statistics for Wind Vectors, Monthly Weather Review, 140, 3204 – 3219, https://doi.org/10.1175/MWR-D-12-00028.1, 2012.

Theuer, F., van Dooren, M. F., von Bremen, L., and Kühn, M.: Minute-scale power forecast of offshore wind turbines using single-Doppler long-range lidar measurements, Wind Energy Science, 5, 1449–1468, https://doi.org/10.5194/wes-5-1449-2020, 2020.

Theuer, F., van Dooren, M. F., von Bremen, L., and Kühn, M.: Lidar-based minute-scale offshore wind speed forecasts analysed under
115  different atmospheric conditions, Meteorologische Zeitschrift, 31, 13–29, https://doi.org/10.1127/metz/2021/1080, 2021.

Würth, I., Ellinghaus, S., Wigger, M., J Niemeier, M., Clifton, A., and W Cheng, P.: Forecasting wind ramps: can long-range lidar increase accuracy?, Journal of Physics: Conference Series, 1102, 012 013, https://doi.org/10.1088/1742-6596/1102/1/012013, 2018.

---

## Author Response (AR2)

**Author's response to the editor comments regarding the paper**

**Observer-based power forecast of individual and aggregated offshore wind turbines**

**Frauke Theuer, Andreas Rott, Jörge Schneemann, Lueder von Bremen, and Martin Kühn**

Dear Prof. Dr. Pryor,

thank you for once again taking the time to review our manuscript and providing valuable comments. We address each of your comments below. Changes to the manuscript are visualized in blue.

*1) Provide a clearer statement of the novelty of the work. (i.e. Anonymous referee 1 first comment). You added some (4 I believe) references to section 2 but still do not offer a clear statement regarding novelty of the findings/research methods etc.*

We have now added more explicit statements regarding the novelty of the work throughout Section 2:

- Line 83f.: The reference method probabilistic lidar-based power forecast (LF) using single lidar measurements was developed by Theuer et al. (2020a) and is based on the work of Valldecabres et al. (2018) who applied dual-Doppler radar.

- Line 119f.: The SCADA-based power forecast (SF) modifies the methodology introduced in Rott et al. (2020), adapting its wind vector weighting approach and time scales to match the LF.

- Line 135f.: In this work we extend the LF to an observer-based power forecast (OF) by integrating the SF.

- Line 141f.: EMOS is commonly used to calibrate ensemble forecasts; in our work it is applied to minute-scale remote sensing-based power forecasts for the first time.

- Line 164f.: In our work we apply the method to a data set with higher temporal resolution and shorter forecast horizons.

*2) Do an analysis of whether your forecast skill is linked to wake impacts (reviewer 1 comment 2). Yes you can remove the impact from the hard targets, but what the reviewer actually asks is does the presence of the upstream wind farm and resulting wakes impact forecast skill. So if you can assess that it would be very interesting.*

Unfortunately we are not able to perform such an analysis as during the time of the campaign/forecast neither Hohe See nor Albatros were operational. Thus, no wind turbine or wind farm wakes were present that could have impacted the forecast. To make this more clear we slightly adapted the sentence mentioning Hohe See and Albatros that we added during the review process (Line 211 ff.):

Figure 1 (a) additionally depicts the layout of the wind farms Albatros and Hohe See, which were under construction but not yet operational during the time of the analysis. Those turbines did not cause any wakes but were only visible as hard targets in the lidar scans occasionally, which were omitted during data filtering and thus did not impact the forecast.

*3) Add text from your response to qu 6 from reviewer 2 into the manuscript*

As we already discuss the impact of forecast horizon on forecast skill in general (Line 501ff.) and the impact of Lagrangian

advection, propagation distances and duration on forecast uncertainty (Line 408ff., Line 502f.) we did not add this part of our response to the discussion. We included a condensed version of the discussion on the range of the lidar, scanning trajectories, wind farm layout and their relation to forecast horizon and skill in Line 403f. As this is already discussed in-depth in Theuer et al. (2020b) and our analysis in this work did not focus on this topic we decided against a more detailed discussion.

Generally, the LF's skill is highest for free flow turbines and areas covered well by the lidar scans. As discussed in more detail in Theuer et al. (2020b), lidar range, scanning trajectory and wind farm layout do not only influence the forecast availability but can also impact forecast uncertainty and relate to e.g. a forecast bias.

Moreover, we included a sentence regarding the expected value of the oberver-based forecast against the benchmark persistence for larger lead times and the further analysis planned in that context in Line 507ff..

[revised manuscript text omitted]